# PARETOROUTER: VLSI GLOBAL ROUTING WITH MULTI-OBJECTIVE OPTIMIZATION

## ABSTRACT

Global routing (GR) has been a central task in modern chip design. Many efforts, either ML-based or heuristic, have been proposed that seek to optimize specific business goals, such as overflow (OF) and wirelength (WL) of generated routes. Notably, recent end-to-end neural routers have demonstrated significant speed advantages in optimizing wirelength, yet they struggle to achieve efficiency in reducing overflow. In fact, a good trade-off between the above two metrics has not been achieved, especially when overall efficiency is pursued, as existing ML-based methods often optimize only a single metric. To bridge this gap for more practical industry applications, we propose a flow-matching-based router for GR, called ParetoRouter, which achieves trade-offs between WL and OF, generating highly connected routes at high speed and quality. In the training phase, two differential metric-oriented routing results are utilized to build the training datasets. They are leveraged to design an 'Average Flow' between initial pins and final routings. A Pareto sampling method, based on the Das-Dennis method, is also devised to achieve trade-offs between OF and WL in the inference phase. Extensive experimental results show that it achieves SOTA performance on the **overflow reduction** with **less superfluous routes** across all benchmarks with **x10** times speedup over the peer SOTA ML-based method.

## 1 INTRODUCTION

Global routing (GR) (McMurchie et al., 1995; Cheng et al., 2022; Liao et al., 2020) has become one of the most intricate and time-consuming phases in modern VLSI design flows (Kramer & Van Leeuwen, 1984), among the other stages such as logic synthesis (Neto et al., 2021), floorplanning (Li et al., 2022a), and placement (Hao et al., 2021; Shi et al., 2023). With netlists containing millions to billions of nets, global routers interconnect pins, aiming to minimize wirelength and avoid overflow under limited routing resources. In fact, even the simplified '2-pin' case, i.e., connecting each net with exactly two pins under specified constraints, is NP-complete (Paulus et al., 2021).

Classical works on heuristics (Cho et al., 2007; Kastner et al., 2002) often require continuous updates and improvements by human experts to route greedily. To mitigate the reliance on manual effort and enhance design automation and quality, learning-based approaches have been introduced. Notably, as shown in Table 1, most existing ML-based works often rely on generative models or classical solvers (Li et al., 2022b; Yan et al., 2018; Li et al., 2021; 2024; Feng & Feng, 2025) to predict Steiner points (Hwang, 1979) and heuristics post-processing or deep reinforcement learning (DRL) (Mahboubi et al., 2021) to route the discrete predicted points, e.g. NeuralSteiner (Liu et al., 2024) and Hubrouter (Du et al., 2023). DSBRouter (Shi et al., 2025) leverages Diffusion Schrödinger Bridge (DSB) (De Bortoli et al., 2021) to design an end-to-end ML-based router (e.g., from discrete initial pins to connected routes), but it is hard to train and suffers a great generation time. In addition, most ML-based methods optimize only a single metric: for example, Hubrouter primarily targets wirelength, whereas NeuralSteiner and DSBRouter focus on overflow, making it challenging to achieve trade-offs between these two metrics.

Motivated by accelerated diffusion sampling paradigms (e.g., Consistency Models (Song et al., 2023), CM, Flow Matching (Lipman et al., 2023), FM), can we design an end-to-end global router that optimizes both overflow and wirelength simultaneously, generating high-quality connected routes while significantly reducing generation time?

Table 1: Summary of existing machine learning-based solvers for GR.

| METHOD | TYPE | END-TO-END | MOO SUPPORT |
|---|---|---|---|
| Hubrouter (Du et al., 2023) | GENERATIVE + RL | ✗ | ✗ |
| NeuralSteiner (Liu et al., 2024) | CNN + POST-PROCESSING | ✗ | ✗ |
| DSBRouter (Shi et al., 2025) | GENERATIVE | ✓ | ✗ |
| ParetoRouter (ours) | GENERATIVE | ✓ | ✓ |

Thus, in this paper, we propose ParetoRouter, which is an FM-based global router that integrates a carefully designed yet simple 'Average Flow' (AF) to fulfill the diversity of predicted routes, facilitating on-demand sampling of routing results. Simply put, during training, ParetoRouter utilizes the combination of two differential metric-oriented connected routes as supervisory signals, enabling the model to predict the AF in latent space. The design of AF is detailed in Sec 4.1. The FM model and the DSB module in DSBRouter serve similar functions.

However, as demonstrated by the experimental results in Appendix A.2.2 and A.6.2, FM achieves comparable performance to DSB at a lower training cost. In the sampling phase, a fast one-step Pareto sampling method based on Das-Dennis approach (Das & Dennis, 1998) is designed to realize a controllable generation of routes, which means that ParetoRouter can generate connected routes that are more inclined towards overflow (or wirelength) to approximate Pareto Front (PF). This sampling module is detailed in Sec. 4.2. Extensive experiments show that ParetoRouter can not only generate routes with diversity, but also significantly reduce the generation time and generate nearly state-of-the-art (SOTA) routing results under rationally specified parameters. **This paper contributes as follows**:

1) To the best of our knowledge, the proposed ParetoRouter is the first ML-based multi-objective global router that produces preference-compliant routing solutions and explicitly explores the Pareto front between wirelength (WL) and overflow (OF).

2) ParetoRouter incorporates a fast Das–Dennis–based Pareto sampling scheme to approximate Pareto Front in GR, which substantially reduces solution-generation time by up to 90% compared with DSBRouter (Shi et al., 2025). It enables on-demand OF-oriented or WL-oriented routing results, compared to SOTA ML-based methods Du et al. (2023); Liu et al. (2024).

3) Experimental results demonstrate that ParetoRouter achieves SOTA OF performance across all evaluated benchmarks, reducing OF by an average of $68\%$ and markedly decreasing superfluous routing on large-scale nets compared to the SOTA ML-based DSBRouter.

## 2 PRELIMINARIES AND PROBLEM DEFINITION

### 2.1 OFFLINE MULTI-OBJECTIVE OPTIMIZATION

Offline multi-objective optimization (MOO) aims to simultaneously minimize multiple objectives using an offline dataset $\mathcal{D}$ of designs and their corresponding labels. Let the design space be $\mathcal{X} \subseteq \mathbb{R}^d$, where $d$ denotes the dimensionality of the design. The goal of MOO is to identify solutions that achieve the best trade-offs among conflicting objectives. Formally, the multi-objective optimization problem is defined as:

$$\text{Find } \boldsymbol{x}^* \in \mathcal{X} \text{ such that there is no } \boldsymbol{x} \in \mathcal{X} \text{ with } f(\boldsymbol{x}) \prec f(\boldsymbol{x}^*), \tag{1}$$

where $f : \mathcal{X} \to \mathbb{R}^m$ is a vector-valued map of $m$ objective functions, and $\prec$ denotes Pareto dominance. A solution $\boldsymbol{x}$ *Pareto dominates* another solution $\boldsymbol{x}^*$ (denoted $f(\boldsymbol{x}) \prec f(\boldsymbol{x}^*)$) if:

$$\forall i \in \{1, \ldots, m\}, \ f_i(\boldsymbol{x}) \leq f_i(\boldsymbol{x}^*) \quad \text{and} \quad \exists j \in \{1, \ldots, m\} \text{ such that } f_j(\boldsymbol{x}) < f_j(\boldsymbol{x}^*). \tag{2}$$

In other words, $\boldsymbol{x}$ is no worse than $\boldsymbol{x}^*$ on every objective and strictly better on at least one. A solution $\boldsymbol{x}^*$ is *Pareto optimal* if there is no $\boldsymbol{x} \subseteq \mathcal{X}$ that Pareto dominates $\boldsymbol{x}^*$. The set of all Pareto-optimal solutions is the *Pareto set (PS)*. The corresponding set of objective vectors, $\{f(\boldsymbol{x}) \mid \boldsymbol{x} \in PS\}$, is known as the *Pareto front (PF)*.

The goal of MOO is to compute a set of solutions that closely approximates the PF, providing a comprehensive representation of the best attainable trade-offs among the objectives.

## 2.2 Flow Matching

Flow matching (FM) (Lipman et al., 2023) is an advanced generative modeling framework that has demonstrated superior effectiveness and efficiency compared to other models (Ho et al., 2020; Song et al., 2021b;a). At its core is a conditional probability path $p_t(\boldsymbol{x} \mid \boldsymbol{x}_0)$ for $t \in [0, 1]$, which evolves from the initial distribution $p_0(\boldsymbol{x} \mid \boldsymbol{x}_0) = q(\boldsymbol{x}_0)$ to an approximate Dirac delta $p_1(\boldsymbol{x} \mid \boldsymbol{x}_0) \approx \delta(\boldsymbol{x} - \boldsymbol{x}_0)$. This evolution is conditioned on a specific point $\boldsymbol{x}_0$ drawn from $q(\boldsymbol{x}_0)$ and is governed by the conditional vector field $u_t(\boldsymbol{x} \mid \boldsymbol{x}_0)$. A neural network with parameters $\boldsymbol{\theta}$ is trained to learn the marginal vector field $v(\boldsymbol{x}, t)$:

$$\hat{v}(\boldsymbol{x}, t; \boldsymbol{\theta}) \approx v(\boldsymbol{x}, t) = \mathbb{E}_{\boldsymbol{x}_0 \sim p_t(\boldsymbol{x}|\boldsymbol{x}_0)}[u_t(\boldsymbol{x} \mid \boldsymbol{x}_0)]. \tag{3}$$

The modeled vector field $\hat{v}(\boldsymbol{x}, t; \boldsymbol{\theta})$ serves as a neural Ordinary Differential Equation (ODE) that guides the transport from $q(\boldsymbol{x}_0)$ to $p_{data}(\boldsymbol{x}_1)$. Normally, FM begins with a sampled noise $\boldsymbol{x}_0$ from $q(\boldsymbol{x}_0)$ (Pooladian et al., 2023). Then a linear interpolation with the uncorrupted data $\boldsymbol{x}_1$ is constructed:

$$\boldsymbol{x} \mid \boldsymbol{x}_0, t = (1 - t) \cdot \boldsymbol{x}_0 + t \cdot \boldsymbol{x}_1, \ \boldsymbol{x}_0 \sim q(\boldsymbol{x}_0). \tag{4}$$

The conditional vector field is readily derived as $u_t(\boldsymbol{x} \mid \boldsymbol{x}_0) = (\boldsymbol{x}_0 - x)/(1 - t)$. Equivalently, $u_t(\boldsymbol{x} \mid \boldsymbol{x}_0) = \boldsymbol{x}_1 - \boldsymbol{x}_0$. The following objective is used to minimize the conditional FM:

$$\mathbb{E}_{t, p_{data}(\boldsymbol{x}_0), q(\boldsymbol{x}_1)} \| \hat{v}(\boldsymbol{x}, t; \boldsymbol{\theta}) - (\boldsymbol{x}_1 - \boldsymbol{x}_0) \|^2 . \tag{5}$$

After training, samples are generated by integrating the neural ODE driven by the learned vector field $\hat{v}(\boldsymbol{x}, t; \boldsymbol{\theta})$.

## 2.3 Global Routing via FM

Typically, given a physical chip and a netlist, a chip canvas is created along with several nets (as shown in Fig. 1 (a)), where each net includes pins placed at fixed positions determined by the placement of macros and standard cells. The primary goal of global routing (GR) is to establish connections for all required pins while simultaneously minimizing the routing WL and OF. Existing ML-based methods (Li et al., 2024; Du et al., 2023; Liu et al., 2024) generally approach GR as a two-phase task (refer to Fig. 1(a), (b), and (c)): first, the Steiner points (pins) are predicted, and then post-processing algorithms are applied to connect the initial pins with the predicted Steiner points. Some approaches, such as DSBRouter (Shi et al., 2025), aim to create an end-to-end pipeline (see Fig. 1(a) and (d)). However, the DSB used in DSBRouter suffers from efficiency problems, and it is unable to sample routes that satisfy multi-objective optimization (MOO).

In contrast, ParetoRouter, depicted in Fig. 1(e), leverages a flow-based model (FM) to learn the distribution $p_{\boldsymbol{\theta}}(\boldsymbol{x}_1|\boldsymbol{x}_0)$ of routes $\boldsymbol{x}_1$ with high diversity for a given instance $\boldsymbol{x}_0$. Design of the AF will be discussed in Sec. 4.1. For sampling, we design a one-step, Das-Dennis-based Pareto sampling method, which accelerates the generation process and leads to routing results approximating the Pareto Front. The design of the sampling method is detailed in Sec. 4.2.

## 3 Related Works

Due to page limits, we leave partial related works to Appendix A.1.

**The Task of Global Routing.** Owing to the complexity of VLSI routing, the circuit layout is partitioned into rectangular regions, called global cells (GCells) (Cho et al., 2007). Global routing is modeled as a grid graph $G = (V, E)$, where each GCell corresponds to a vertex ($v \in V$) and adjacent GCells are connected by an edge ($e \in E$) representing their shared boundary. Modern chip designs employ two or more metal layers for routing, where each layer is assigned either a horizontal or vertical direction, yielding a two-dimensional grid abstraction. For each net, the global router assigns a connected subset of GCells—linked via multiple edges—to interconnect all pins, typically producing a rectilinear Steiner tree (RST) (Chu & Wong, 2005). The Hanan grid (Hanan, 1966) and the escape graph (Ganley & Cohoon, 1994) are often exploited to construct a shortest rectilinear Steiner minimum tree (RSMT) while avoiding obstacles (Liu et al., 2012), by treating intersection points in these graphs as candidate Steiner points.

**Classical Global Router.** Global routing is a combinatorial optimization problem that can be formulated as a 0–1 integer linear program and solved with a general-purpose solver. In practice,

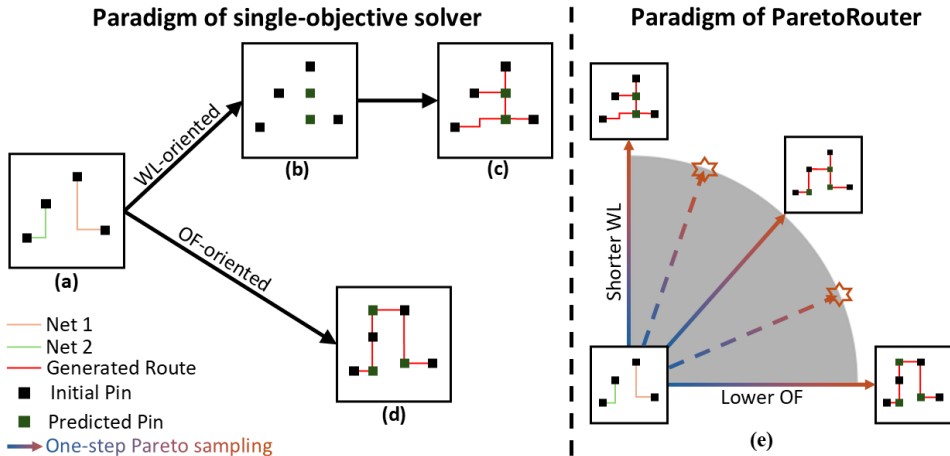

Figure 1: **Difference between ParetoRouter and other solvers.** (a): initial pins and nets. (b): predicted pins and initial pins. (c) and (d): Generated routes via post-processing algorithms. (e): One-step Pareto sampling within ParetoRouter.

classical routers decompose the task into two stages aimed at congestion management: Steiner topology generation and rip-up-and-reroute (RRR). The former commonly relies on FLUTE (Chu & Wong, 2005), which uses lookup tables to build near-optimal Steiner trees in terms of WL for each net, but it is oblivious to congestion. During congestion mitigation, many routers perform edge shifting to move routing demand out of congested regions (Chu & Wong, 2005). NTHU-Route 2.0 (Chang et al., 2008) further introduces a history-based cost that accumulates past congestion and dynamically adjusts routing costs, improving overall quality and efficiency. NCTU-GR 2.0 (Liu et al., 2013) adopts an SMT-aware routing scheme to achieve shorter WL. However, as design complexity and scale grow, these procedures become increasingly time-consuming. Consequently, accelerating congestion resolution with deep learning–based techniques can improve the overall performance of global routing.

**Learning-based Router.** A growing body of works investigates learning to optimize WL and the use of neural networks for global routing, including generating pin-connection orders (Liao et al., 2020), routing segments (Cheng et al., 2022), and customized hub points for rectilinear Steiner trees (RSTs) (Du et al., 2023; Li et al., 2024; Feng & Feng, 2025). The primary practical challenges, however, lie in handling large-scale nets to mitigate overflow (OF) and maintain short WL under limited routing resources. In such settings, judicious detours are essential for relieving congestion, because the WL-minimal RST, e.g., that in Fig. 1(c) produced by Hubrouter (Du et al., 2023) or NeuralSteiner (Liu et al., 2024), may be infeasible in practice. DSBRouter (Shi et al., 2025) can produce low-OF solutions (Fig. 1(d)), but often introduces excessive redundant routing and incurs prohibitive generation time on large-scale nets. PatLabor (Chen et al., 2025b) considers the MOO constraint, but its focus is on balancing WL and RSMT construction delay. Moreover, global routers must be able to generalize to unseen circuit distributions. To meet these challenges, we propose *ParetoRouter*, which enables explicit trade-offs between OF and WL, yielding on-demand routing solutions for nets of arbitrary scale.

## 4 FRAMEWORK OF PARETOROUTER

For ML-based global routing (GR) solvers, the two-stage paradigm of first predicting Steiner points and then performing routing is intuitive and straightforward. However, it exhibits several limitations: (i) the neural networks used to predict Steiner points are typically trained in a supervised manner, making them brittle under distribution shift and prone to inflating the time complexity of downstream post-processing due to prediction noise; (ii) relying on a single classical solver to provide training targets restricts the diversity of Steiner points available to the model; and (iii) post-processing algorithms are often focused primarily on WL (or OF) minimization, which makes it challenging to satisfy multi-objective optimization (MOO) constraints.

To address these issues, this section introduces ParetoRouter, which integrates MOO into GR to handle constraints explicitly and perform GR in an end-to-end manner. The next two sections first present AF, a training-time mechanism that leverages routing results from multiple classical solvers

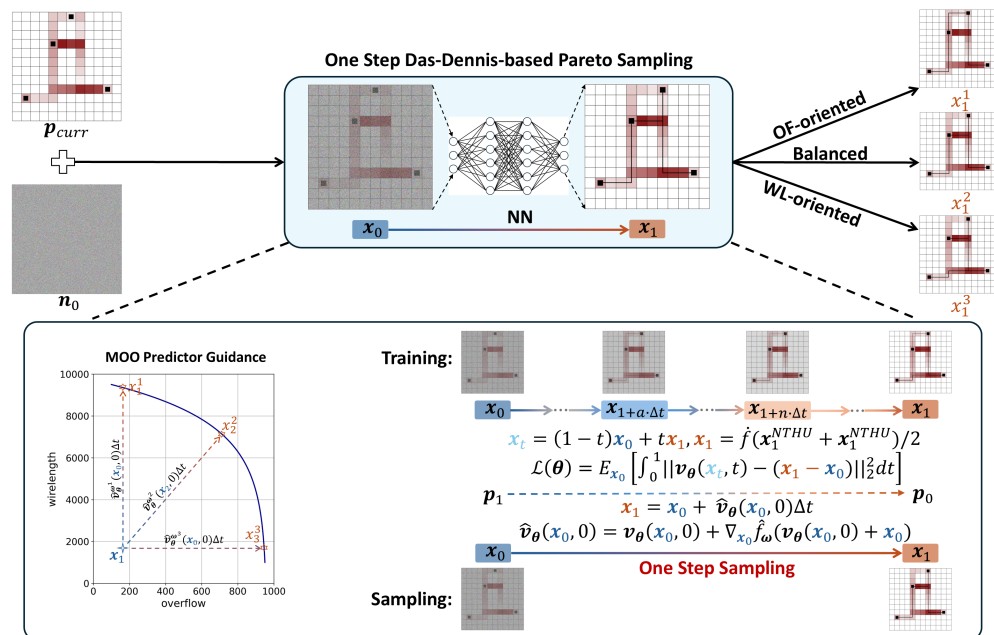

Figure 2: For training, ParetoRouter applies the average of two different solvers' routing results (NTHURouter and NCTU-GR) to construct the supervisory signals $x_0$ to enable the NN module to predict more potential routes. While in the sampling phase, weights $\omega = \{\omega^i\}$ subdivide the objective space into equal partitions. Each weight $\omega^i \in \omega$ maps to a sampled routing result $x_0^i$.

to enable the model to predict a diverse set of routes. Then, during sampling, we introduce a one-step Pareto sampling pipeline based on the Das–Dennis method to initialize multiple weight matrices, thereby guiding the rapid generation of MOO-compliant routes under different weightings in a single step. Fig. 2 depicts the pipeline of proposed ParetoRouter.

## 4.1 AVERAGE FLOW

Diffusion solvers for GR are not something new; both Hubrouter (Du et al., 2023) and DSBRouter (Shi et al., 2025) try to fulfill the potential of diffusion models for GR. Hubrouter leverages GAN (Goodfellow et al., 2014), DDPM (Ho et al., 2020), and VAE (Kingma & Welling, 2013) to sample predicted 'hubs' from simulated Gaussian noise. Anchoring the generative process to stationary noise to enable standard sampling paradigms is intuitive but exhibiting a disconnect between the diffusion trajectory and the structured nature of solution spaces, i.e., the generative process operates in the Gaussian noise space for pin prediction rather than the solution (routes), which causing the restriction on both the controllability of the intermediate states within the generation and the exploitation of prior pin knowledge and need of post-processing algorithms for connected routes. In contrast, DSBRouter (Shi et al., 2025) designs a post-processing-free paradigm that better aligns with heuristic search dynamics, but it is hard to train the backbone and takes a great time for generation due to the inherent constraints of DSB (De Bortoli et al., 2021). Besides, all the ML-based solvers discussed above optimize either WL or OF; none of them can generate routes according to the realistic needs. Consequently, ParetoRouter introduces AF as:

$$\mathbb{E}_{t,p_{data}(x_1),q(x_0)} \| \hat{v}(x, t; \theta) - (\dot{f}(x_1^{NTHU} + x_1^{NCTU})/2 - x_0) \|^2, \tag{6}$$

where $x_1^{NTHU}$ and $x_1^{NCTU}$ represent the routing results of NTHURouter (Chang et al., 2008) and NCTU-GR (Liu et al., 2013), respectively. $\dot{f}$ is a scale function to scale the differential routes of these two solvers. $x_0$ is the noisy pins with congestion map. We introduce Gaussian noise $n_0$ to corrupt the initial pins $p_{curr}$, resulting in an initial sample $x_0 = f_n(n_0 + p_{curr})$ drawn from $q(x_0)$, where $f_n$ represents the normalization function. Reasons for noise corruption will be discussed in Appendix A.2.1. Given two flows, starting with the same $x_0$, ending in $x_1^{NTHU}$ and $x_1^{NCTU}$ respectively, the loss design is like predicting the flow in between. This is why we name it Average Flow. This design is straightforward but has been proven to be very effective in the experiments, which will be discussed in Sec. 5.

### 4.2 ONE STEP DAS-DENNIS-BASED PARETO SAMPLING

This section first elucidates the concept of gradient guidance under multi-objective optimization (MOO) constraints within the ParetoRouter sampling pipeline, along with the design. Then, the detailed formulation of a weighted score distribution induced by the Das–Dennis method is introduced to fulfill guided Pareto sampling.

**Sampling with Gradient Guidance.** Classifier guidance was initially introduced to steer sample generation toward designated image categories (Dhariwal & Nichol, 2021). This concept has since been extended to regression contexts, where it is employed to guide molecular generation (Lee et al., 2023; Jian et al., 2024; Chen et al., 2025a). Building upon Lemma 1 from Zheng et al. (2023), we derive the formulation of gradient guidance within the framework of flow matching as follows:

$$\tilde{v}(\boldsymbol{x}, t; \boldsymbol{\theta}) = \hat{v}(\boldsymbol{x}, t; \boldsymbol{\theta}) + \rho \cdot \nabla_{\boldsymbol{x}_t} \log p(\boldsymbol{s} \mid h(\boldsymbol{x}_t, t)), \tag{7}$$

where $p(s \mid \boldsymbol{x}_t, t)$ denotes the distribution of predicted routing score and $\boldsymbol{s}$ denotes the computed properties through classifier function $h$ whose implementation will be detailed in Sec. A.2.2. More details of Eq.(7) can be found in Appendix A.3. In implementation, as we assume a one-step sampling procedure, where the NN module within the ParetoRouter framework is designed to predict the difference between $\boldsymbol{x}_1$ and $\boldsymbol{x}_0$. Consequently, we adopt the following formulation to better align with this one-step sampling scheme:

$$\tilde{v}(\boldsymbol{x}, t; \boldsymbol{\theta}) = \hat{v}(\boldsymbol{x}, t; \boldsymbol{\theta}) + \rho \cdot \nabla_{\boldsymbol{x}} \log p(\boldsymbol{s} \mid h(\hat{v}(\boldsymbol{x}, t; \boldsymbol{\theta}) + \boldsymbol{x}_0, t)). \tag{8}$$

**Weighted Score Distribution.** Preceding ML-based works like DSBRouter primarily address the generation of samples that satisfy a single score $s$. In contrast, our proposed ParetoRouter is designed to optimize two properties, namely OF and WL simultaneously, represented as $\boldsymbol{s} = [\hat{f}_1(\boldsymbol{x}_t), \hat{f}_2(\boldsymbol{x}_t)]$. To address the increased complexity inherent in this multi-objective setting, we decompose the overall generation task into a series of weighted single-objective subproblems. Specifically, we introduce a weight vector $\boldsymbol{\omega} = [\omega_1, \omega_2]$, where each $\omega_i > 0$ and $\sum_{i=1}^{m} \gamma_i \omega_i = 1$. The resulting weighted score is:

$$\hat{f}_{\boldsymbol{\omega}}(\boldsymbol{x}_t) = -h\left(\sum_{i=1}^{n} \gamma_i \hat{f}_i(\boldsymbol{x}_t)\omega_i\right). \tag{9}$$

Here, $\hat{f}_i$ denotes the predicted score of the $i$th objective for $\boldsymbol{x}_t$. Given that the scales of OF and WL differ, a scaling factor $\boldsymbol{\gamma}$ is introduced to ensure compatibility. The negative sign reflects that the objective is minimization. Following the approach in Lee et al. (2023), we define the weighted score distribution:

$$p(\boldsymbol{s} \mid \hat{v}(\boldsymbol{x}, t; \boldsymbol{\theta}) + \boldsymbol{x}_0, t) = e^{\hat{f}_{\boldsymbol{\omega}}(\hat{v}(\boldsymbol{x}, t; \boldsymbol{\theta}) + \boldsymbol{x}_0)} / Z, \tag{10}$$

where $Z$ is the normalization constant. By incorporating this formulation into Eq.8, we arrive at the following predictor update:

$$\tilde{v}(\boldsymbol{x}, t; \boldsymbol{\theta}) = \hat{v}(\boldsymbol{x}, t; \boldsymbol{\theta}) + \rho \cdot \nabla_{\boldsymbol{x}} \hat{f}_{\boldsymbol{\omega}}(\hat{v}(\boldsymbol{x}, t; \boldsymbol{\theta}) + \boldsymbol{x}_0). \tag{11}$$

This vector field $\tilde{v}(\boldsymbol{x}, t; \boldsymbol{\theta})$ effectively guides the sampling process toward regions in the input space that satisfy the desired multi-objective properties encoded by the weighted distribution. To achieve comprehensive coverage of the Pareto objective space, we employ the Das-Dennis approach (Das & Dennis, 1998), which partitions the objective space uniformly to generate a diverse set of weight vectors $\boldsymbol{\omega}$. Each weight vector corresponds to a distinct sampled route, thereby facilitating exploration across the entire trade-off front. The sampling step is performed using the Euler method (Van Kampen, 1976), formulated as:

$$\hat{\boldsymbol{x}}_j = \boldsymbol{x}_t + \tilde{v}(\boldsymbol{x}, t; \boldsymbol{\theta})\Delta t, \tag{12}$$

where $j = t + \Delta t$ represents the next time step. In the ParetoRouter framework, we set $t = 0$ and $\Delta t = 1$. The complete procedure is summarized in Algorithm 1.

## 5 EXPERIMENTS

In this section, we empirically compare our proposed ParetoRouter with other ML-based and classical solvers on ISPD benchmarks.

---

**Algorithm 1 One-Step Pareto Sampling**

---

**Input:** Dataset $\mathcal{D}$, epochs $T$;
**Output:** Generated connected router set $\mathcal{R}$;
  1: Train the vector field $\hat{v}(\boldsymbol{x}, t; \boldsymbol{\theta})$ of FM using loss from Eq.(6) on $\mathcal{D}$.
  2: Generate weight vectors $\{\boldsymbol{\omega}^i\}_{i=1}^{N}$ using the Das-Dennis method.
  3: Initialize $\boldsymbol{x}_1$ with scaled Gaussian noise $\boldsymbol{n}$ using two classical GR solvers.
  4: **for** $t = 1$ to $T$ **do**
  5:    Set $\Delta t = 1$
  6:    Calculate the score distribution using Eq.(10).
  7:    Calculate the guided vector field $\tilde{v}(\boldsymbol{x}, t; \boldsymbol{\theta})$ using Eq.(11).
  8:    Derive sampled routes (Pareto Front) with Eq.(12).
  9: **end for**
10: **return** $\mathcal{R}$

---

Table 2: **Crrt, WLR and generation time on ISPD07 benchmarks.** Note GAN-HubRouter can not directly produces a fully connected result, while DSBRouter consumes a great generation time. Compared with them, our ParetoRouter achieves 100% correctness rate with a small generation time.

| METRIC | CASE | HUBROUTER (GAN) (DU ET AL., 2023) | DSBROUTER (SHI ET AL., 2025) | PARETOROUTER (OURS) |
|---|---|---|---|---|
| **Correctness Rate** | SMALL-4 | $0.48 \pm 0.004$ | $\mathbf{1.000 \pm 0.000}$ | $\mathbf{1.000 \pm 0.000}$ |
| | SMALL | $0.12 \pm 0.001$ | $\mathbf{1.000 \pm 0.000}$ | $\mathbf{1.000 \pm 0.000}$ |
| | LARGE-4 | $0.000 \pm 0.000$ | $\mathbf{1.000 \pm 0.000}$ | $\mathbf{1.000 \pm 0.000}$ |
| | LARGE | $0.000 \pm 0.000$ | $\mathbf{1.000 \pm 0.000}$ | $\mathbf{1.000 \pm 0.000}$ |
| **Wirelength Ratio** | SMALL-4 | $\mathbf{1.012 \pm 0.011}$ | $1.015 \pm 0.000$ | $1.016 \pm 0.000$ |
| | SMALL | $1.002 \pm 0.001$ | $\mathbf{1.001 \pm 0.000}$ | $1.002 \pm 0.000$ |
| | LARGE-4 | $1.004 \pm 0.021$ | $\mathbf{1.001 \pm 0.000}$ | $1.002 \pm 0.000$ |
| | LARGE | $\mathbf{1.001 \pm 0.000}$ | $1.002 \pm 0.000$ | $1.003 \pm 0.000$ |
| **Generation Time (GPU Sec)** | SMALL-4 | $\mathbf{5.88 \pm 0.11}$ | $2643 \pm 3.11$ | $7.66 \pm 0.30$ |
| | SMALL | $7.15 \pm 0.09$ | $2671 \pm 1.68$ | $\mathbf{8.92 \pm 0.06}$ |
| | LARGE-4 | $\mathbf{6.00 \pm 0.07}$ | $2687 \pm 3.30$ | $8.27 \pm 0.19$ |
| | LARGE | $\mathbf{7.82 \pm 0.10}$ | $2571 \pm 2.24$ | $12.33 \pm 0.11$ |

## 5.1 SETTINGS

For evaluation, we conduct experiments on both ISPD07 (newblue04–newblue07 and adaptec01–adaptec05) and ISPD98 (ibm01–ibm06) benchmarks (Alpert, 1998). For both benchmarks, we use WL, OF, and generation time as the primary evaluation criteria. Our proposed ParetoRouter is compared with three classical routing algorithms — GeoSteiner (Juhl et al., 2018), Labyrinth (Kastner et al., 2002), FlUTE (Wong & Chu, 2008) and ES (Chu & Wong, 2005) — as well as three SOTA ML-based methods: Hubrouter (Du et al., 2023), NeuralSteiner (Liu et al., 2024) and DSBRouter (Shi et al., 2025). We also stduy the Correctness Rate (Crrt), Wirelength Ratio (WLR) (Cheng et al., 2022) and Generation Time on newblue04–newblue07 as DSBRouter does.

It is worth noting that four SOTA solvers (Liu et al., 2024; Feng & Feng, 2025; Chen et al., 2025b; Li et al., 2024) are either tailored to benchmarks with different standards (Liang et al., 2024; Dolgov et al., 2019) or have not been publicly released (Liu et al., 2024; Chen et al., 2025b; Feng & Feng, 2025). Because NeuralSteiner evaluates on the same benchmarks as the two open-source solvers (Hubrouter and DSBRouter), we report its results as provided in the original paper to enable a fair comparison. Further details on the experimental benchmarks and additional supplementary experiments are given in Appendix A.5.2; A.6.1.

## 5.2 CRRT AND WLR ON PARTIAL ISPD07 BENCHMARKS

Following prior ML-based studies Shi et al. (2025), we evaluate ParetoRouter against existing ML-based solvers on the ISPD07 benchmarks (newblue04–newblue07), partitioned into four categories—small, small-4, large, and large-4—consistent with earlier works. For simplicity, we report results only for GAN-Hubrouter, as other Hubrouter variants yield less competitive results. We also cancel RL-based post-processing for GAN-Hubrouter to ensure a fair comparison (Shi et al., 2025). As shown in Table 2, ParetoRouter and DSBRouter, as two end-to-end solvers, achieve routing solutions with 100% connectivity, whereas Hubrouter without post-processing maintains

Table 3: **Wirelength (WL) & overflow (OF) on ISPD98:** classical global routing (GeoSteiner, Labyrinth, Flute+RES) and ML-based methods (Hubrouter, NeuralSteiner, DSBRouter).

| Metrics | Model | ibm01 | ibm02 | ibm03 | ibm04 | ibm05 | ibm06 |
|---------|-------|-------|-------|-------|-------|-------|-------|
| WL | GeoSteiner | **60142** | **165863** | **145678** | **162734** | **409709** | **275868** |
| | Labyrinth | 75909 | 201286 | 187345 | 195856 | 420581 | 341618 |
| | Flute+ES[*] | 61492 | 169251 | 146287 | 167547 | 411936 | 280477 |
| | HR-VAE | $64703 \pm 1498$ | $176492 \pm 6830$ | $159968 \pm 3281$ | $179895 \pm 5274$ | $434942 \pm 2916$ | $301144 \pm 5832$ |
| | HR-DPM | $66464 \pm 1586$ | $190588 \pm 2337$ | $168454 \pm 2486$ | $183696 \pm 1736$ | $475820 \pm 5516$ | $320423 \pm 2958$ |
| | HR-GAN | $61056 \pm 151$ | $167545 \pm 236$ | $147050 \pm 208$ | $164298 \pm 326$ | $411857 \pm 472$ | $277977 \pm 514$ |
| | NeuralSteiner[*] | 61735 | 170405 | 148036 | 166648 | 415684 | 283727 |
| | DSBRouter | 61435 | 174016 | 152862 | 163942 | 420464 | 342349 |
| | ParetoRouter (ours) | 63386 | 174896 | 155993 | 171859 | 482016 | 307696 |
| OF | GeoSteiner | 3342 | 7399 | 3944 | 7420 | 401 | 8033 |
| | Labyrinth | **292** | 384 | 122 | 1124 | **0** | 502 |
| | Flute+ES[*] | 3100 | 7121 | 3699 | 6889 | 317 | 7821 |
| | HR-VAE | $4721 \pm 667$ | $9919 \pm 801$ | $7311 \pm 692$ | $10433 \pm 1299$ | $909 \pm 106$ | $14103 \pm 1684$ |
| | HR-DPM | $4933 \pm 700$ | $14117 \pm 1309$ | $9344 \pm 818$ | $11471 \pm 871$ | $2390 \pm 126$ | $17229 \pm 1500$ |
| | HR-GAN | $3491 \pm 64$ | $7481 \pm 31$ | $4010 \pm 42$ | $7551 \pm 22$ | $419 \pm 7$ | $8039 \pm 12$ |
| | NeuralSteiner | 2200 | 3800 | 2100 | 2700 | 18 | 2833 |
| | DSBRouter | 1430 | 0 | 4 | 10 | 0 | 11858 |
| | ParetoRouter (ours) | 1051 | **0** | **0** | **0** | **0** | **0** |
| TIME | GeoSteiner | **3.08** | 6.91 | 6.80 | **9.07** | **7.72** | **7.66** |
| | Labyrinth[*] | 7.11 | 11.08 | 11.61 | 42.03 | 12.70 | 21.02 |
| | Flute+ES[*] | 3.14 | **4.90** | **5.88** | 15.49 | 7.88 | 14.11 |
| | HR-VAE | $9.66 \pm 0.08$ | $9.69 \pm 0.04$ | $10.19 \pm 0.06$ | $12.93 \pm 0.07$ | $14.58 \pm 0.00$ | $17.28 \pm 0.16$ |
| | HR-DPM | $1796.09 \pm 38.68$ | $2772.29 \pm 16.83$ | $2936.52 \pm 21.23$ | $3865.21 \pm 25.07$ | $4369.47 \pm 22.56$ | $4965.08 \pm 121.46$ |
| | HR-GAN | $41.02 \pm 0.51$ | $46.58 \pm 0.56$ | $52.04 \pm 2.35$ | $67.31 \pm 3.51$ | $72.28 \pm 3.72$ | $88.02 \pm 4.45$ |
| | NeuralSteiner[*] | 27.18 | 34.79 | 46.24 | 50.37 | 75.99 | 70.32 |
| | DSBRouter | 4991 | 5667 | 8418 | 10745 | 11313 | 11858 |
| | ParetoRouter (ours) | 42.37 | 61.96 | 67.96 | 68.09 | 91.20 | 131.80 | 121.76 |

[*] Experimental results cited from raw manuscripts.

only about 30% connectivity on average. ParetoRouter preserves WLR performance comparable to both Hubrouter and DSBRouter, while delivering generation time on par with Hubrouter and superior to DSBRouter. Unless otherwise noted, all reported ParetoRouter results were obtained with $\omega_1 = \omega_2 = 0.5$.

## 5.3 OF and WLR on real-world benchmarks

Noting that, as ParetoRouter produces multiple results to satisfy the MOO constraints and OF is relatively important than WL (Liu et al., 2024), we present the WL and OF of the sampled routes that reduce OF the most.

**Routing Results on ISPD98.** Table 3 shows the WL, OF and generation time for all tested methods on ISPD98 benchmarks. For OF, ParetoRouter achieves the most OF reduction. Compared with the SOTA ML-based OF-oriented DSBRouter, ParetoRouter significantly reduces the total OF with a reduction of 36.06% on ibm01 and 100% on ibm05 and only uses an average 1/10 generation time of DSBRouter. In terms of wirelength, ParetoRouter does not incur too much loss, with the least 5.24% on ibm01 and the most 15.00% on ibm05 compared with GeoSteiner. For generation time, ParetoRouter remains at the same level as NeuralSteiner, but there is still a gap compared to VAE-based Hubrouter and other classical methods.

**Routing Results on ISPD07.** Table 4 shows the WL, OF and generation time for selected tested methods on ISPD07 benchmarks. We keep the GAN-based Hubrouter and skip other variants of Hubrouter, as the GAN variant gets the best outcome. With the size of nets increasing, ParetoRouter and DSBRouter get the most OF reduction compared to all other methods across all tested benchmarks. But, compared to DSBRouter, ParetoRouter does not incur much increase in WL. For WL and generation time, ParetoRouter shows a similar performance compared to ISPD98.

## 5.4 Ablation Study

Series of ablation studies are conducted to study the effectiveness of classifier gradient guidance, proposed loss function, NN module, as well as classifier guidance module.

**Role of Loss Function, NN Module, and Classifier Guidance Module.** To assess the roles of these three components within the ParetoRouter framework, we perform ablations separately. For the loss function, we remove the $x_1^{NTHU}$ term. For the NN module and the classifier-gradient guidance module,

Table 5: OF & WL w/ varying ablated components on ibm01.

| Loss | OF | WL | Time |
|------|-----|-----|------|
| w/o $x_1^{NCTU}$ | 1211 | 63771 | 41.11 |
| w/o $x_1^{NTHU}$ | 1379 | 63529 | 42.56 |
| w/o NN | 1565 | 63450 | 27.07 |
| w/o Guidance | - | - | - |
| ParetoRouter | **1051** | **63386** | 42.37 |

we remove the corresponding components from the model architecture. We report results for OF and

Table 4: **Wirelength (WL) & overflow (OF) on ISPD07.** Comparison of 2 selected classical global routing (GeoSteiner, Flute+RES) and 3 ML-based methods (Hubrouter, NeuralSteiner, DSBRouter).

| METRICS | MODEL | ADAPTEC01 | ADAPTEC02 | ADAPTEC03 | ADAPTEC04 | ADAPTEC05 |
|---|---|---|---|---|---|---|
| WL | GEOSTEINER | **3389601** | **3209172** | **9330748** | **8865643** | **9784471** |
|  | NCTU-GR | 3623718 | 3331725 | 9598156 | 9087206 | 10560933 |
|  | NTHUROUTER* | 5344000 | 5229000 | 13101000 | 12169000 | 15538000 |
|  | FLUTE+ES* | 3418461 | 3235803 | 9417934 | 8896007 | 9886249 |
|  | HR-GAN | 3407033 | 3229110 | 9355980 | 8888775 | 9832110 |
|  | NEURALSTEINER* | 3438717 | 3247429 | 9459117 | 9003952 | 9915795 |
|  | DSBROUTER | 12299050 | 10072054 | 29478326 | 24276147 | - |
|  | PARETOROUTER(OURS) | 3522091 | 3386722 | 9502199 | 9021491 | 10288329 |
| OF | GEOSTEINER | 35945 | 53848 | 142254 | 45050 | 102300 |
|  | NCTU-GR | **0** | **0** | **0** | **0** | **0** |
|  | NTHUROUTER* | **0** | **0** | **0** | **0** | **0** |
|  | FLUTE+ES* | 32518 | 50947 | 137104 | 42306 | 957704 |
|  | HR-GAN | 35441 | 53652 | 142131 | 45230 | 102108 |
|  | NEURALSTEINER* | 82 | 255 | 728 | 97 | 431 |
|  | DSBROUTER | **0** | **0** | **0** | **0** | - |
|  | PARETOROUTER(OURS) | **0** | **0** | **0** | **0** | **0** |
| TIME | GEOSTEINER | 92.70 | 123.00 | 371.02 | 311.19 | 320.07 |
|  | NCTU-GR | **8.96** | 8.31 | 26.67 | 21.00 | 27.45 |
|  | NTHUROUTER* | 10.0 | **2.10** | **10.90** | **2.60** | **23.00** |
|  | FLUTE+ES* | 118.48 | 187.03 | 396.51 | 376.72 | 360.68 |
|  | HR-GAN | 593.02 | 780.44 | 1324.81 | 1387.01 | 1384.96 |
|  | NEURALSTEINER* | 347.20 | 461.35 | 1351.91 | 1138.66 | 1106.54 |
|  | DSBROUTER | 65624 | 119353 | 115438 | 125589 | - |
|  | PARETOROUTER(OURS) | 422.20 | 469.51 | 1561.30 | 1418.09 | 1500.11 |

*EXPERIMENTAL RESULTS CITED FROM RAW MANUSCRIPTS.

WL on `ibm01`. As ParetoRouter cannot guarantee connectivity of the generated routes without the guidance module, we thus don't report results of the NN-ablated ParetoRouter. Table 5 shows that both WL and OF are affected by all three components. When $x_1^{NTHU}$ is ablated, both OF and WL increase compared to the complete ParetoRouter. When the guidance module is ablated, OF increases markedly, whereas WL decreases, which is reasonable since Geostiner emphasizes WL optimization. Taken together, these results demonstrate the effectiveness of the proposed components within the ParetoRouter framework.

**Influence of $\omega$.** To evaluate whether the proposed Das–Dennis–based sampling can effectively manage the Pareto Front, we examine the generated weightings $\gamma_i \cdot \omega_i$. The performance variation observed across different weightings underscores the effectiveness of the proposed sampling scheme in controlling the Pareto front. Specifically, we generate 10 uniformly spaced weightings to approximate the Pareto front in GR. For illustration, we select four distinct $\omega$ for ParetoRouter and conduct experiments on `ibm01`, as `ibm01` exhibits greater variability

Table 6: OF & WL w/ varying $\gamma_i \cdot \omega_i$ on ibm01.

| $\gamma_1 \cdot \omega_1$ | $\gamma_2 \cdot \omega_1$ | OF | WL |
|---|---|---|---|
| 0 | 1 | 1505 | **63317** |
| 0.2 | 0.8 | **1051** | 63386 |
| 0.5 | 0.5 | 1493 | 63563 |
| 0.8 | 0.2 | 1671 | 63847 |
| 1 | 0 | 1682 | 63956 |

in OF. As shown in Table 6, as the ratio $\gamma_1 \cdot \omega_1$ increases from 0 to 0.2, OF reaches its minimum at $\gamma_1 \cdot \omega_1 = 0.2$ and then increases as $\gamma_1 \cdot \omega_1$ continues to grow, whereas WL increases monotonically. This pattern suggests that generating more routes (i.e., larger WL) can exacerbate congestion along existing routes. These observations also provide indirect evidence for the effectiveness of the proposed guided sampling in managing the Pareto Front.

## 6 CONCLUSION AND OUTLOOK

In this paper, we introduce *ParetoRouter*, an end-to-end ML-based global router. Equipped with a simple AF loss and a classifier-gradient guidance module subject to multi-objective optimization (MOO) constraints, ParetoRouter can generate either OF-oriented or WL-oriented routes with $100\%$ connectivity for previously unseen large-scale nets in a single step. Experimental results show that ParetoRouter reduces overflow by an average of $68\%$ while incurring only a modest wirelength penalty—particularly on large-scale nets, thereby narrowing the gap between ML-based solvers and practical chip-design applications.

**Limitations and Future Work.** However, the NN module can still predict superfluous routes that are incorporated into the final routing solution, increasing WL. Moreover, compared with some ML-based solvers (e.g., GAN-Hubrouter and NeuralSteiner), ParetoRouter exhibits slightly longer generation times, which we attribute to the additional gradient computations. In future work, we will focus on optimizing generation time and further reducing WL.

ETHICS STATEMENT

This paper aims to advance the state of the art in machine learning and artificial intelligence for electronic design automation (AI4EDA). This paper does not present immediate, direct negative social impacts. While the research may entail various societal implications, we do not identify any that warrant specific emphasis in this paper.

REPRODUCIBILITY STATEMENT

To ensure the reproducibility of our research, description of our methodology is detailed comprehensively, so as the implementation and experimental setups. All experimental results in the paper are reproducible, and the implementation code of ParetoRouter/code for reproducing experimental results will be fully open sourced on Github upon publication of this paper.

LLM USAGE STATEMENT

The contribution of Large Language Models (LLMs) in the work presented in this article is limited to: 1. polishing the given written statements; 2. reviewing the syntax of the written sentences. We declare that no experimental results, core implementation of our search, core scientific ideas, experimental designs, or conclusions have been generated or modified by LLMs. The LLM we used is GPT-5, owned by OpenAI, and no other LLMs were utilized. All authors have reviewed the final version of the manuscript and take full responsibility for its content and originality.

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

---

**Algorithm 2 Training of FM within ParetoRouter**

---

**Input:** Dataset $\mathcal{D}$, epochs $epos$, Time scheduler $T$, NthuRouter (Chang et al., 2008) $NTHU$,
    NCTU-GR (Liu et al., 2013) $NCTU$;
**Output:** Vector field $\hat{v}(\boldsymbol{x}, t; \boldsymbol{\theta})$;
 1: Initialize model parameters $\boldsymbol{\theta}$.
 2: Construct $q(x_0)$ and $p_{data}(x_1)$ utilizing $\mathcal{D}$.
 3: **for** $epo = 1$ to $epos$ **do**
 4:     Sample a batch of $\boldsymbol{p}_{curr}$ from $q(x_0)$.
 5:     Sample a Gaussian noise $\boldsymbol{n}_0$ like $\boldsymbol{p}_{curr}$.
 6:     Construct $\boldsymbol{x}_0$ utilizing $\boldsymbol{n}_0$ and $\boldsymbol{p}_{curr}$ under $\boldsymbol{x}_0 = f_n(\boldsymbol{n}_0 + \boldsymbol{p}_{curr})$.
 7:     Derive $\boldsymbol{x}_1^{NTHU}$ and $\boldsymbol{x}_1^{NCTU}$ from $NTHU(\boldsymbol{p}_{curr})$ and $NCTU(\boldsymbol{p}_{curr})$, respectively.
 8:     Sample $t$ from $T$.
 9:     Derive $\boldsymbol{x}_t$ using Eq. 4.
10:     Derive output of the vector field $\hat{v}(\boldsymbol{x}, t; \boldsymbol{\theta})$.
11:     Compute loss $\mathcal{L} = \mathbb{E}_{t, p_{data}(\boldsymbol{x}_0), q(\boldsymbol{x}_1)}$ using Eq. 6.
12:     Loss.backward().
13: **end for**
14: **return** $\hat{v}(\boldsymbol{x}, t; \boldsymbol{\theta})$.

---

# A APPENDIX

## A.1 SUPPLEMENTAL RELATED WORKS

The section reviews works on offline multi-objective optimization and guided generative modeling.

**Offline Multi-Objective Optimization (MOO).** Most MOO research has focused on the online setting, where a black-box function is queried interactively to optimize multiple objectives simultaneously (Jiang et al., 2023; Park et al., 2023; Gruver et al., 2023). By contrast, offline MOO is often more realistic because online queries may be costly or risky (Xue et al., 2024). In the offline regime, a learned predictor serves as the oracle and enables two classical families of methods. Evolutionary algorithms conduct population-based search via iterative parent selection, reproduction, and survivor selection (Zhang et al., 2021; Li, 2024; Yuan et al., 2025). Bayesian optimization instead leverages the predictor within an acquisition function to select promising candidates, iteratively refining the search through sampled evaluations (Daulton et al., 2023; Golovin & Zhang, 2020; Qing et al., 2023). Training the predictor can be further improved by techniques such as COMs (Trabucco et al., 2021), ROMA (Yu et al., 2021), NEMO (Fu & Levine, 2021), ICT (Yuan et al., 2023), Tri-mentoring (Chen et al., 2023), GradNorm (Chen et al., 2018), and PcGrad (Yu et al., 2020), which enhance training efficiency and stability. **Guided Generative Modeling.** A parallel line of work develops generative models that produce samples meeting multiple desired properties. For example, Wang et al. (2021) incorporates structure–property relations into a conditional Transformer to bias generation, and Wang et al. (2022) employs a VAE to recover semantics and property correlations by modeling weights in the latent space. Tagasovska et al. (2022) apply multiple-gradient descent to trained EBMs to synthesize new samples, though training an EBM per property is complex. Han et al. (2023) explores a distinct setting aimed at generating modules that satisfy specified conditions. Zhu et al. (2023) use GFlowNets as acquisition functions, and Jain et al. (2023) integrate multiple objectives into GFlowNets. Yao et al. (2024) induce diversity via hand-crafted penalties rather than uniform weight vectors in a white-box setting. Gruver et al. (2023) investigate online multi-objective optimization within a diffusion framework, using an acquisition function to guide sampling, while Kong et al. (2024) apply multi-objective guidance under diffusion but assume equal weights across properties, which cannot recover the Pareto front. Related work on guided diffusion also targets single-objective optimization (Chen et al., 2024; Yuan et al., 2024). Overall, many of these approaches rely on generators that are either less expressive or difficult to train De Bortoli et al. (2021). In contrast, ParetoRouter pairs a SOTA flow-matching model with classifier-gradient guidance for sampling.

---

**Algorithm 3 Classifier Function $h$**

---

**Input:** Vector field $\hat{v}(\boldsymbol{x}, t; \boldsymbol{\theta})$, $\boldsymbol{x}_0$, scaling factors $\boldsymbol{\gamma}$;
**Output:** Weighted score $\hat{f}_{\boldsymbol{\omega}}(\boldsymbol{x}_t)$;
 1: Derive weightings $\boldsymbol{\omega}$ leveraging Das-Dennis approach (Das & Dennis, 1998).
 2: Derive sampled $\boldsymbol{x}_1'$ using the vector field $\hat{v}(\boldsymbol{x}, 0; \boldsymbol{\theta})$ and $\boldsymbol{x}_0$.
 3: Extract predicted routing map $\boldsymbol{r}$ from $\boldsymbol{x}_1'$.
 4: Compute complete routing map $\boldsymbol{r}'$ under minimal $\sum_{i=1}^{n} \gamma_i \hat{f}_i(\boldsymbol{x}_1')\omega_i$.
 5: Require $\boldsymbol{x}_1'$.gradient.
 6: Initialize mask map $\boldsymbol{m} = 0$.
 7: **for** $r \in \mathbb{R}' \cup \mathbb{R}$ **do**
 8:     **if** $r \in \mathbb{R} \wedge r \in \mathbb{R}'$ **then**
 9:         pass
10:     **else if** $r \in \mathbb{R} \wedge r \notin \mathbb{R}'$ **then**
11:         mask map $\boldsymbol{m}(r) = 1$.
12:     **else** $r \notin \mathbb{R} \wedge r \in \mathbb{R}'$
13:         mask map $\boldsymbol{m}(r) = -1$.
14:     **end if**
15: **end for**
16: $h(\hat{v}(\boldsymbol{x}, t; \boldsymbol{\theta}) + \boldsymbol{x}_0) = \boldsymbol{r} \cdot \boldsymbol{m}$
17: Compute $\hat{f}_{\boldsymbol{\omega}}(\boldsymbol{x}_t)$ utilizing Eq. 9.
18: **return** $\hat{f}_{\boldsymbol{\omega}}(\boldsymbol{x}_t)$.

---

### A.2 Supplemental Algorithms

This section introduces the training algorithms in Algorithm. 1 (Line 1) and classifier function $h$ in Eq. 7.

#### A.2.1 Training of FM

Training of the FM within ParetoRouter is summarized in Algorithm 2. We inject scaled noise $\boldsymbol{n}_0$ into the clean data for the following reasons: Through experiments, we find that the intensity of the injected noise has a negligible effect on both FM training and Das–Dennis sampling. However, without this perturbation the backbone converges poorly (i.e., the trained FM module cannot reliably compute, under supervision, the flow that bridges the initial pins $\boldsymbol{x}_0$ and the final connected routes $\boldsymbol{x}_1$.) We therefore conclude that adding appropriately scaled noise to clean data facilitates the model training. In practice, we sample Gaussian noise, add it to the clean $\boldsymbol{p}_{curr}$, and then normalize the corrupted $\boldsymbol{x}_0$ using the normalization function $f_n$.

#### A.2.2 Classifier Function

ParetoRouter employs a guidance module, first introduced in DSBRouter (Shi et al., 2025), to steer route generation, as shown in Algorithm 3. Nevertheless, there are essential differences between the guidance used by ParetoRouter and that in DSBRouter.

Firstly, DSBRouter applies an SDE-based gradient guidance (Li et al., 2023) to drive DSB generation, whereas ParetoRouter operates within the FM framework. DSBRouter adopts SDSB (Tang et al., 2024) as its backbone and leverages the series proposed theories in Tang et al. (2024) together with the energy-function formalism (LeCun et al., 2006) to justify an SDE-based guidance of the form:

$$p_{\boldsymbol{\theta}}(x_t \mid x_{t+1}, \boldsymbol{g}^*) = Z p_{\boldsymbol{\theta}}(\boldsymbol{x}_t \mid \boldsymbol{x}_{t+1}) p(\boldsymbol{g}^* \mid \boldsymbol{x}_t) \tag{13}$$

where $\boldsymbol{g}^*$ denotes the optimal objective score. Since ParetoRouter uses Eq. 7 within the FM framework to guide the generation process, the underlying working principles are different, and the DSBRouter guidance cannot be directly applied to ParetoRouter.

Secondly, ParetoRouter integrates multi-objective optimization (MOO) constraints into the design of its classifier-guidance module, whereas DSBRouter considers only the reduction of the objective

function (OF). In DSBRouter, the following formula:

$$\nabla_{\boldsymbol{x}_{t+1}} \log p(\boldsymbol{g}^* | \boldsymbol{x}_{t+1}) = \nabla_{\boldsymbol{x}_{t+1}} (E^o_{\boldsymbol{x}_{t+1} \sim p_r(\boldsymbol{x}_{t+1}|\boldsymbol{x}_{t+2})}(\eta(\boldsymbol{x}_{t+1})) - O(\eta(\boldsymbol{x}_{t+1}))) \tag{14}$$

is used to approximate $p(\boldsymbol{g}^* \mid \boldsymbol{x}_t) \propto \exp([\nabla_{\boldsymbol{x}_{t+1}}(E^o_{\boldsymbol{x}_{t+1} \sim p_r(\boldsymbol{x}_{t+1}|\boldsymbol{x}_{t+2})}(\eta(\boldsymbol{x}_{t+1})) - O(\eta(\boldsymbol{x}_{t+1})))]^\intercal x_t)$. Because the gradient of $O(\eta(\boldsymbol{x}_{t+1}))$ is zero, Eq. 14 effectively uses the gradient of $E^o_{\boldsymbol{x}_{t+1} \sim p_r(\boldsymbol{x}_{t+1}|\boldsymbol{x}_{t+2})}(\eta(\boldsymbol{x}_{t+1}))$ to compute $\nabla_{\boldsymbol{x}_{t+1}} \log p(\boldsymbol{g}^* | \boldsymbol{x}_{t+1})$. Here, $E^o$ in DSBRouter is the expected routing given $\boldsymbol{x}_{t+2}$, and $E$ is computed under the constraint:

$$\arg \min_{\bar{\boldsymbol{x}}_t} |E^o_{\bar{\boldsymbol{x}}_t \sim p_r(\bar{\boldsymbol{x}}_t|\boldsymbol{x}_{t+1})}(\eta(\bar{\boldsymbol{x}}_t)) - O(\eta(\bar{\boldsymbol{x}}_t)) + c(\bar{\boldsymbol{x}}_t)| \tag{15}$$

which indicates that DSBRouter aims solely to minimize the OF. In contrast, ParetoRouter employs a neural-network–free classifier that explicitly accounts for MOO constraints:

$$\arg \min_{\boldsymbol{x}'_t} |\sum_{i=1}^n \gamma_i \hat{f}_i(\boldsymbol{x}'_t)\omega_i| \tag{16}$$

This design aligns with the objective of ParetoRouter's generation process and enables more diverse routing results compared with DSBRouter (Shi et al., 2025).

## A.3 DERIVATION OF EQ. 7

This section derives Eq. 7. By Lemma 1 in Zheng et al. (2023), the guided vector field takes the form:

$$\tilde{v}(\boldsymbol{x}, t; \boldsymbol{\theta}) = a_t \boldsymbol{x}_t + b_t \nabla_{\boldsymbol{x}} \log p(\boldsymbol{x}_t \mid s) \tag{17}$$

where $a_t = \frac{\dot{a}_t}{a_t}$ and $b_t = (\dot{a}_t \sigma_t - a_t \dot{\sigma}_t)\frac{\sigma_t}{a_t}$. Setting $a_t = t$ and $\sigma_t = 1 - t$, Eq. 17 simplifies to:

$$\tilde{v}(\boldsymbol{x}, t; \boldsymbol{\theta}) = \frac{1}{t}\boldsymbol{x}_t + \frac{1-t}{t}\nabla_{\boldsymbol{x}} \log p(\boldsymbol{x}_t \mid s) \tag{18}$$

The conditional log-probability function is written as

$$\log p(\boldsymbol{x}_t \mid y) = \log p_{\boldsymbol{\theta}}(\boldsymbol{x}_t) + \log p(s \mid h(\boldsymbol{x}_t, t)) - \log p(s) \tag{19}$$

where $p_{\boldsymbol{\theta}}(\boldsymbol{x}_t)$ denotes the data distribution learned by the flow matching model and $p(s \mid h(\boldsymbol{x}_t, t))$ is the classified property distribution.

Substituting these expressions yields

$$\tilde{v}(\boldsymbol{x}, t, y; \boldsymbol{\theta}) = \frac{1}{t}\boldsymbol{x}_t + \frac{1-t}{t}\nabla_{\boldsymbol{x}} \log p_{\boldsymbol{\theta}}(\boldsymbol{x}_t) + \frac{1-t}{t}\nabla_{\boldsymbol{x}_t} \log p(s \mid h(\boldsymbol{x}_t, t)) \tag{20}$$

$$= \tilde{v}(\boldsymbol{x}, t; \boldsymbol{\theta}) + \frac{1-t}{t}\nabla_{\boldsymbol{x}_t} \log p(s \mid h(\boldsymbol{x}_t, t)) \tag{21}$$

## A.4 PARETOROUTER NETWORK ARCHITECTURE

### A.4.1 BACKBONE

We adopt a symmetric U-Net with time-conditioned residual blocks and attention. Starting from 64×64 RGB inputs, a 7×7 stem (3→64) feeds four encoder stages with channel widths [64, 64, 128, 256] and a 512-channel bottleneck. Each encoder stage contains two ResNet blocks with adaptive instance normalization (AdaIN) modulated by a learned time embedding, a self-attention module (linear attention in the first three stages and full attention at the deepest stage), and a 2× downsampling operation, producing the resolution sequence 64→32→16→8→4. Attention uses four heads with 32-dimensional subspaces and four learnable memory key–value pairs. At 4×4, the bottleneck expands into two hyperconnected residual streams, applies two 512-channel ResNet blocks interleaved with full attention, and then reduces back to a single stream. The decoder mirrors the encoder: at each resolution it concatenates the corresponding skip features (doubling the channel dimensionality), applies two ResNet blocks with 1×1 residual projections, inserts attention (full at the first decoder stage, linear thereafter), and upsamples via nearest-neighbor interpolation followed by a convolution. RMSNorm is used throughout, linear attention is employed at higher resolutions to

Table 7: **Summary of the test dataset.** We respectively show the scale size, verticalhorizontal capacity, number of nets, and average/maximum number of pins for each net.

| CASE | IBM01 | IBM02 | IBM03 | IBM04 | IBM05 | IBM06 | ADA01 | ADA02 | ADA03 | ADA04 | ADA05 |
|---|---|---|---|---|---|---|---|---|---|---|---|
| SIZE | $64 \times 64$ | $80 \times 64$ | $80 \times 64$ | $96 \times 64$ | $128 \times 64$ | $128 \times 64$ | $324 \times 324$ | $424 \times 424$ | $774 \times 779$ | $774 \times 779$ | $465 \times 468$ |
| CAP.(V/H) | 24/28 | 44/68 | 40/60 | 40/46 | 84/126 | 40/66 | 70/70 | 80/80 | 62/62 | 62/62 | 110/110 |
| NETS | 11507 | 18429 | 21621 | 26163 | 27777 | 33354 | 219794 | 260159 | 466295 | 515304 | 867441 |
| AVG.PINS | 4.31 | 4.88 | 4.10 | 3.86 | 5.25 | 4.21 | 4.29 | 4.09 | 4.02 | 3.71 | 4.03 |
| MAX.PINS | 42 | 134 | 55 | 46 | 17 | 35 | 2271 | 1935 | 3713 | 3974 | 9863 |

reduce complexity from O(n²) to O(n), while full attention is retained at the lowest resolution. The overall downsampling factor is 16, so input height and width must be divisible by 16.

Temporal conditioning is provided by a 64-D sinusoidal positional encoding passed through a two-layer GELU MLP to produce a 256-D time embedding, in each ResNet block, AdaIN applies feature-wise scaling and shifting derived from this embedding, i.e., norm(x)·(scale+1)+shift. Under a conventional instantiation with two 3×3 convolutions per residual block, 1×1 projections where required, 128-D attention projections (4×32 heads), and the above time-conditioning MLP, the model comprises approximately 21.5 million trainable parameters: 20.9M in convolutional/residual pathways, 0.79M in attention projections, and 0.08M in the time-embedding MLP, with normalization parameters being negligible.

### A.4.2 MOEL PARAMETERS

For training, learning rate $lr$ is fixed to $0.0003$ and batch size is set to $256$. Training of FM is under fp16 precision. For sampling, the sampling steps is set to 1, aligned with the description in 4.2.

### A.4.3 REASONS FOR CHOOSING NCTU-GR AND NTHUROUTER

Previously, single-objective ML-based solvers typically used the solutions produced by a traditional solver as supervisory signals. For example, HubRouter Du et al. (2023) is supervised using NCTU-GR (Liu et al., 2013), whereas DSBRouter Shi et al. (2025) is supervised using NTHURouter (Chang et al., 2008). In our initial experiments, we also used only the outputs of NTHURouter as the supervisory signal and observed performance comparable to DSBRouter. This naturally raises the question of whether combining the outputs of multiple solvers as a weighted supervisory signal can enlarge the effective search space explored by the CFG-guided reverse (denoising) diffusion process, thereby improving the results of multi-objective optimization. Motivated by this hypothesis, we incorporate supervision from two empirically strong solvers, NTHURouter and NCTU-GR. Our ablation study confirms the effectiveness of the proposed loss design. However, although we do not include additional solver outputs, we argue that the benefit of such supervision is unlikely to grow linearly with the number of solvers. A larger effective search space implies a longer guided generation trajectory, and as the search space grows without bound, the additional exploration becomes almost indistinguishable from the noise injected at time $t_0$, ultimately degrading the model's performance.

## A.5 EXPERIMENTAL PROTOCOLS

### A.5.1 DATASETS AND HARDWARE FOR EXPERIMENTS

We evaluate our approach on the real-world datasets ISPD07 (Nam et al., 2007) and ISPD98 (Alpert, 1998). Following Du et al. (2023); Shi et al. (2025); Liu et al. (2024), we build low-overflow expert training datasets by using Nthurouter (Chang et al., 2008) and low-wirelength expert training datasets by using NCTU-GR (Liu et al., 2013) to route a subset of the ISPD07 benchmarks—bigblue1, bigblue2, bigblue3, bigblue4, newblue4, newblue5, newblue6, and newblue7. The dataset pre-processing follows the pipeline outlined in Hubrouter (Du et al., 2023). Interaction of the two solvers' routing results are selected, resulting in nearly 220k samples in total. We initialize the capacities as specified by the benchmarks and route the nets sequentially using the results of Chang et al. (2008). After each capacity update, we generate a condition image comprising the current capacity and the pin locations for the next net, and we simultaneously produce and store the corresponding ground-truth route image. Both images are randomly clipped, when feasible, to a common resolution of $64 \times 64$. For the evaluation reported in Table. 2, we select newblue1, newblue2 from ISPD07—outside the training set—with a total of 10k samples. A summary of the ISPD98 test cases is provided in Table. 7. We prepare the ISPD98 test cases using the same processing pipeline as in Du et al. (2023).

Table 8: **Relative error on ISPD98.** The routing results of Juhl et al. (2018) are treated as the theoretical lower bound. Optimal results are in bold.

| MODEL | IBM01 | IBM02 | IBM03 | IBM04 | IBM05 | IBM06 |
|---|---|---|---|---|---|---|
| LOWER BOUND | 60142 | 165863 | 145678 | 162734 | 409709 | 275868 |
| LABYRINTH | 0.262 | 0.213 | 0.286 | 0.203 | 0.026 | 0.238 |
| FLUTE+ES | 0.023 | 0.017 | **0.007** | 0.027 | 0.007 | 0.016 |
| HR-VAE | $0.075 \pm 0.024$ | $0.064 \pm 0.04$ | $0.098 \pm 0.022$ | $0.105 \pm 0.032$ | $0.061 \pm 0.007$ | $0.091 \pm 0.021$ |
| HR-DPM | $0.105 \pm 0.026$ | $0.149 \pm 0.014$ | $0.156 \pm 0.017$ | $0.128 \pm 0.010$ | $0.161 \pm 0.013$ | $0.161 \pm 0.010$ |
| HR-GAN | $0.022 \pm 0.002$ | $\mathbf{0.010 \pm 0.001}$ | $0.009 \pm 0.001$ | $0.009 \pm 0.002$ | $\mathbf{0.005 \pm 0.001}$ | $\mathbf{0.007 \pm 0.001}$ |
| NEURALSTEINER | 0.026 | 0.027 | 0.016 | 0.024 | 0.014 | 0.028 |
| DSBROUTER | **0.021** | 0.049 | 0.049 | **0.007** | 0.026 | 0.240 |
| PARETOROUTER | 0.053 | 0.054 | 0.070 | 0.056 | 0.179 | 0.115 |

Table 9: **Relative error on ISPD07.** The routing results of Juhl et al. (2018) are treated as the theoretical lower bound. Optimal results are in bold.

| MODEL | ADAPTEC01 | ADAPTEC02 | ADAPTEC03 | ADAPTEC04 | ADAPTEC05 |
|---|---|---|---|---|---|
| LOWER BOUND | 3389601 | 3209172 | 9330748 | 8865643 | 9784471 |
| FLUTE+ES | 0.008 | 0.008 | 0.009 | 0.003 | 0.010 |
| HR-GAN | **0.005** | **0.006** | **0.002** | **0.002** | **0.004** |
| NEURALSTEINER | 0.014 | 0.011 | 0.013 | 0.015 | 0.013 |
| DSBROUTER | 2.628 | 2.138 | - | 2.159 | 1.738 |
| PARETOROUTER | 0.039 | 0.055 | 0.018 | 0.017 | 0.051 |

Training of FM and all subsequent experiments are conducted on a machine equipped with an Intel Xeon Platinum 8558 CPU, 8 NVIDIA H200 GPUs (143 GB memory each), and 1600 GB of RAM.

### A.5.2 BASELINES

The baselines referred in Table. 3 are introduced as follows:

1) *GeoSteiner* (Juhl et al., 2018): An optimal RSMT construction solver which get results with SOTA WL.

2) *Labyrinth* (Kastner et al., 2002): A classical routing algorithm that explores how the concept of pattern routing can be utilized to guide the router toward a solution that minimizes interconnect delay while preserving the routability of the circuit.

3) *FLUTE* (Wong & Chu, 2008): A fast and accurate RSMT construction method using a look-up table. It is important to note that this approach can achieve the optimal solution for nets with up to 9 degrees.

4) *Edge Shifting* (Chu & Wong, 2005): A fast, practical RSMT-based algorithm that leverages a specialized lookup table for small nets and a refined recursive splitting approach for larger nets.

5) *Hubrouter* (Du et al., 2023): A global router for RST construction based on reinforcement learning. The hub is generated using a diffusion model, followed by reinforcement learning for RST construction.

6) *NeuralSteiner* (Liu et al., 2024): A two-stage global router. The candidate points are predicted by an RCCA-enhanced CNN, and routing is performed by an RST construction algorithm based on a greedy strategy.

7) *DSBRouter* (Shi et al., 2025): An end-to-end global router based on Diffusion Schrödinger Bridge (DSB) which reach SOTA OF reduction, but is behind in generation time.

### A.6 MORE DISCUSSIONS ABOUT TESTED BASELINES.

In this section, we discuss the related error (Du et al., 2023) of selected tested methods and convergence of proposed training of FM within ParetoRouter.

### A.6.1 RELATED ERROR ON ISPD98 AND ISPD07 CASES

To assess improvements relative to the optimal wirelength rather than absolute values, we compare the relative error on ISPD98 and ISPD07 across all tested methods (Table 8 and Table 9). The relative error is defined as $(WL - LB)/LB$, where $LB$ denotes the theoretical lower bound. On the ISPD98 benchmarks, ParetoRouter produces slightly more superfluous routes than other SOTA ML-based

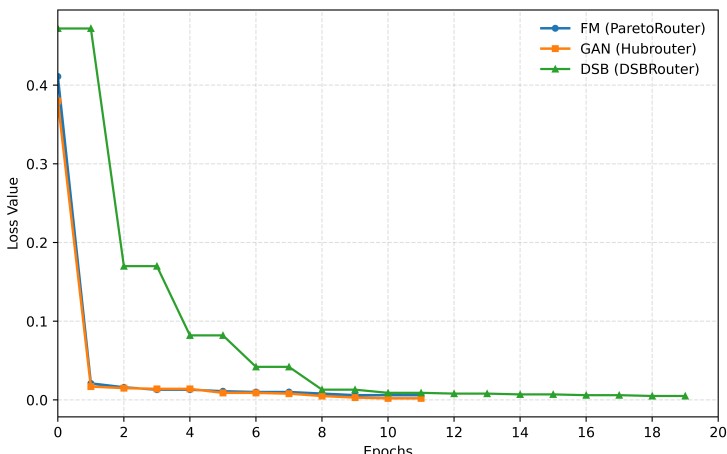

Figure 3: **Convergence of Backbones within different solvers.** Blue line, orange line and green line denote training of FM within ParetoRouter, GAN-based Hubrouter and DSB in DSBRouter.

methods (i.e., DSBRouter (Shi et al., 2025), Hubrouter (Du et al., 2023), NeuralSteiner (Liu et al., 2024)). On the ISPD07 benchmarks, GAN-Hubrouter leads, and ParetoRouter eliminates many superfluous routes compared with DSBRouter, but it still lags behind NeuralSteiner, indicating remaining room for improvement.

### A.6.2 CONVERGENCE OF TRAINING OF FM

To study the convergence of the proposed AF and evaluate the training cost of ParetoRouter, we compare the loss variation during training process across Hubrouter (Du et al., 2023), DSBRouter (Shi et al., 2025) and our proposed ParetoRouter. It needs to be declare that NeuralSteiner (Liu et al., 2024) still disclose the implementation details, so NeuralSteiner is not considered. As shown in Fig. 3, Hubrouter and ParetoRouter can reach convergence in nearly 10 epochs, however, DSBRouter needs 20 epochs to get similar convergence due to alternating training of forward and backward models. This reveals that ParetoRouter can reduce training cost significantly compared to the SOTA ML-based DSBRouter.

### A.7 ADDITIONAL RESULTS

Generated routing results of ParetoRouter across different net scales, along with initial pins, real routing results (ground truth) are shown in Fig. 4.

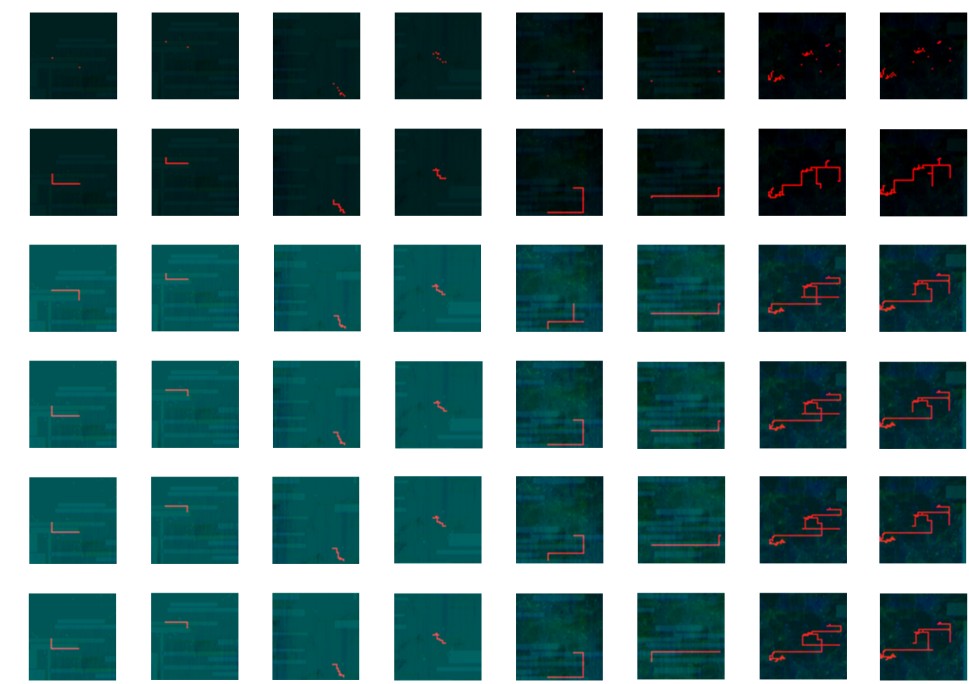

Figure 4: **Examples.** Initial pins $p_{curr}$ (first line), real routing results (second line), generated routing results of ParetoRouter with varying $\gamma_1 \cdot \omega_1 = 0, 0.2, 0.5, 1$ (third line - sixth line), respectively.

