# OpenReview forum: "ParetoRouter: VLSI Global Routing with Multi-Objective Optimization"
_ICLR.cc/2026/Conference — Submitted to ICLR 2026_

### Official Review · Reviewer_bZ4V · 2025-10-28

**Soundness:** 2
**Presentation:** 2
**Contribution:** 2
**Rating:** 4
**Confidence:** 3

**Summary:**

This paper introduces **ParetoRouter**, a novel machine learning-based global routing (GR) method for VLSI design that explicitly addresses the multi-objective optimization (MOO) problem of minimizing both **wirelength (WL)** and **overflow (OF)**. Unlike existing ML-based routers that typically optimize only one metric (e.g., WL-focused HubRouter or OF-focused DSBRouter), ParetoRouter uses a **Flow Matching (FM)** framework to learn a diverse distribution of routing solutions. Experiments on ISPD98 and ISPD07 benchmarks show that ParetoRouter achieves state-of-the-art OF reduction (e.g., 68% average reduction vs. DSBRouter) with competitive WL, while being ~10× faster than DSBRouter and ensuring 100% connectivity.

**Strengths:**

- **Efficiency**: Significant speedup (10×) over DSBRouter due to one-step sampling and FM framework.
- **High Connectivity**: Generates 100% connected routes, unlike non-end-to-end methods like HubRouter.
- **Theoretical Foundation**: Well-grounded in FM and MOO principles, with clear algorithmic design.

**Weaknesses:**

- **Generalization Concerns**: Evaluated on standard benchmarks, but lacks testing on diverse or real-world industrial designs to confirm robustness.
- **Complexity**: Relies on multiple components (AF loss, gradient guidance), which may complicate reproducibility and tuning.
- **Computational Cost**: Slower than some ML-based methods (e.g., VAE-HubRouter) due to gradient computations during sampling.

**Questions:**

1. **Practical Relevance**: How do WL and OF metrics translate to actual business outcomes (e.g., power consumption, timing closure, yield)? Are there results linking ParetoRouter’s outputs to higher-level design goals?
2. **Task Difficulty**: The paper categorizes nets by size (e.g., "large"), but does not define "difficulty" beyond scale. How does ParetoRouter perform on nets with complex congestion patterns or high pin density?
3. **Overfitting and Generalization**: The model is trained on classical solver data. How does it generalize to unseen benchmarks or modern designs with advanced nodes? Were cross-validation or holdout tests performed?
4. **Baseline Comparison**: Why not compare with more recent ML-based methods (e.g., PatLabor) or industrial tools? Are the chosen baselines sufficient to claim SOTA?
5. **Hyperparameter Sensitivity**: How sensitive is performance to the choice of weights (e.g., γ, ω) and guidance strength (ρ)? Is manual tuning required for each design?

---

### Official Review · Reviewer_Kuvv · 2025-10-30

**Soundness:** 2
**Presentation:** 3
**Contribution:** 2
**Rating:** 2
**Confidence:** 3

**Summary:**

This paper designs a flow-matching-based router for global routing, called ParetoRouter, to achieve trade-offs between WL and OF. ParetoRouter is trained on data set with two differential metric-oriented routing results. A Pareto sampling method based on the Das-Dennis method is used to achieve trade-offs between OF and WL in the inference phase.

**Strengths:**

1 ParetoRouter attempts to leverage flow match to produce end-to-end routing solutions

2 ParetoRouter incorporates a fast Das–Dennis–based Pareto sampling scheme, which can reduce solution-generation time.

3. ParetoRouter illustrates its advantage in the experimental study

**Weaknesses:**

1: Some important related works are missed

2: By considering the missed related work, the novelty and contribution should be reconsidered.

**Questions:**

1: One major limitation is lack of the similar work on the combination in the pareto and flow matching, such like “ParetoFlow: Guided Flows in Multi-Objective Optimization ” in ICLR 2025.

2: From ParetoFlow above, we can see that most of the idea of the flow matching and the combination with Pareto Sampling have been proposed. The contribution in this papr should be reconsidered.

3. The proposed methods have different training objective from that of ParetoFlow. The experimental study is needed to illustrate the advantage.

4. It is better to discuss the impact of the relationship between two objectives. This paper mainly performs study on OF and WL. What is about other two objectives with different internal relationships?

5. From table 6, we can see that the adjustment of parameter cannot achieve the desired goal. The weight [0,1] and [1,0] are not always corresponding the best result for OL and WL separately.

6. It is better to discuss the scalability of the method with more than 2 objective functions

---

> ### Author Response · Authors · 2025-11-20
> **Response to reviewer Kuvv**
>
> Thanks for your suggestions.
>
> For the concern about similarity between PareRouter and ParetoFlow, we generally response as following:
>
> We thank the reviewer for pointing out the connection to ParetoFlow. While we indeed build upon the general ideas of flow matching and Pareto-style weighting, we believe our method introduces several substantial and task-specific innovations that are not covered by ParetoFlow:
>
> 1) Problem setting and goal are fundamentally different: ParetoFlow is a general offline MOO algorithm that learns a generative model over abstract design spaces and is evaluated on Off-MOO-Bench (synthetic functions, MO-NAS, MORL, molecule/protein design, portfolio, etc.). Besides, it only expands existing solutions, not a solver. In contrast, ParetoRouter is the multi-objective global router for VLSI, targeting the very specific combination of overflow (OF) and wirelength (WL) under grid-graph capacity constraints, and providing end-to-end nets-to-routes generation with strong speed and quality guarantees on ISPD benchmarks. Designing an MOO algorithm that works in this highly structured, discrete, capacity-constrained routing space requires new modeling and loss design that do not appear in ParetoFlow.
>
> 2) Training objective are different in nature: ParetoFlow uses the standard conditional flow matching loss with linear interpolation and learns a generic data distribution without using any domain-specific solvers. While ParetoRouter, instead, introduces a new Average Flow (AF) loss (Eq. (6) in our paper), which trains the flow to follow the average trajectory between two classical routers. This is a very different use of flow matching: we do not simply model the empirical distribution of routes, but explicitly distill complementary behaviors of two heuristic solvers into a single flow in a way tailored to GR.
>
> 3) Form of Pareto guidance and “Pareto sampling” are architecturally different: ParetoRouter uses a one-step, task-specific Pareto sampling scheme: a). we design a weighted score distribution over OF and WL with explicit rescaling factors to handle the very different scales of the two metrics and a router-specific scoring function $h(\cdot)$; b). ParetoRouter performs one-step Das–Dennis-based sampling rather than multi-step ODE integration with neighbor evolution.
>
> For weakness 1 and 2 (Question 1 and 2), we have added the paretoflow into our supplemental related works (Appendix A.1). Besides, we also revised the contributions declared in Introduction.
>
> For question 3, ParetoRouter has different training objective compared to ParetoFlow due to the fundamental difference of problems solved, so we think there is no advantage needs to be emphasized.
>
> For question 4 and 6, we claim that in task of global (initial) routing, OF and WL are two main evaluated metrics. We explain the two metrics as following:
>
> OF (Congestion or Overflow): Indicates that the line density in certain areas of the chip layout exceeds the designed capacity limit. An OF that is too high means that the circuit layout in the circuit diagram will cause additional delay, energy consumption, and even design failure. Therefore, reducing OF is a crucial goal in routing problems.
>
> WL (Wire Length): Refers to the total length of all connecting lines in the layout. The longer the WL, the higher the manufacturing cost, and the delay in signal transmission will also increase. In short, optimizing WL is mainly aimed at reducing the area of the layout and improving the performance of the circuit.
>
> There is a natural trade-off between these two goals: in order to reduce OF, it is often necessary to take detours or change the level of wiring, which can lead to an increase in WL; On the contrary, reducing WL usually leads to denser wiring, which may cause OF problems. Therefore, OF and WL have become the most classic and common multi-objective optimization problems in VLSI routing. Other metrics like timing and power consumption are less important and hard to integrated into the design of guidance module. And if you wish, we can include more formal analysis of metrics used in VLSI design into appendix.
>
> For question 5, We thank the reviewer for carefully examining Table 6. We believe there is a misunderstanding of the goal of this ablation. The purpose of Table 6 is not to claim that the extreme weights $[1,0]$ and $[0,1]$ must coincide with the globally best OF-only and WL-only solutions, but to show that varying the weight $\gamma_i \cdot \omega_i$ effectively steers the trade-off between OF and WL. As described in Sec. 5.4, Table 6 is an ablation on ibm01 that illustrates how different scalarizations move the solution along the Pareto trade-off, rather than an attempt to recover the single-objective global optima.

---

### Official Review · Reviewer_fSwF · 2025-10-30

**Soundness:** 2
**Presentation:** 3
**Contribution:** 2
**Rating:** 2
**Confidence:** 5

**Summary:**

The authors proposed a flow-matching-based router for GR to trade off WL and OF and achieve better results than the SOTA ML-based router. Their method also shows significant speedup compared with the SOTA ML-based router. However, the contribution of this work is mainly restricted to the ML-based global routing, considering the insights and experiments, which are explained in weakness.

**Strengths:**

1. The authors propose an ML-based global router jointly optimizing overflow and wirelength, which is much better than the SOTA ML-based router, DSBrouter.
2. Their Pareto sampling method provides a large speedup over DSBRouter, which is also based on a generative ML method.

**Weaknesses:**

1. Jointly optimizing overflow (OF) and wirelength (WL) is common sense in global routing. Considering that global routing is a classic problem in the EDA area, I don't think this insight could be an important contribution.
2. The comparison with classic methods is too weak to show the capability of their method. Their mentioned works, e.g., DGR, NTHURouter, and NCTU-GR, are all better global routers than GeoSteiner and FLUTE+ES. Meanwhile, DGR, NTHURouter, and NCTU-GR, all consider overflow and wirelength when formulating and solving the global routing problem. Meanwhile, the results about "LABYRINTH" are missing in Table 3 ("OF") and Table 4 (all metrics).
3. The survey of current works has errors. DGR (Li et al. 2024) and (Feng & Feng 2025) are not ML-based methods. DRG formulated the global routing as a continuous optimization problem and solved it using the deep learning toolkit, PyTorch, for automatic differentiation. Meanwhile, the objective in DGR includes congestion and wirelength cost, which should be classified as a multi-objective method. Therefore, the authors listed them with explicitly wrong information in Table 1.
4. The absence of a crucial ablation study. Only the results "w/o x_1^{NCTU}" are listed in Table 5. In this paper, the average flow is explained in Section 4.1 and Figure 2 in detail as an innovation, so a complete ablation study should be conducted.

**Questions:**

1. In Section 5.4, the author said "For the loss function, we remove the x^{NTHU}_{1} term.". However, the results "w/o x_1^{NCTU}" are listed in Table 5. This may be a typo in the paper, which should be corrected.
2. In Table 5, why are the losses without data from another router (NTHU or NCTU) missing? I wonder if the necessity of incorporating two routers exists? In Section 4.1, the author could explain more about why NTHURouter and NCTU-GR are used for generating a data sample by experimental results.
3. In Table 3, why are the results of LABYRINTH missing under the "OF" metric?
4. In Table 4, why are the results of LABYRINTH missing under all metrics?

---

> ### Author Response · Authors · 2025-11-20
> **Response to reviewer fSwF**
>
> Thanks for your suggestions.
>
> For weakness 1, we also believe that optimizing both WL and OF simultaneously in routing tasks is a common sense, and it seems that there is a problem with the wording of the sentences in our manuscript. What we want to emphasize is that ParetoRouter can generate routing results that meet actual usage preferences, even if there is a trade-off between OF and WL, rather than deliberately emphasizing that ParetoRouter is designed to optimize both WL and OF. Therefore, we have redeclared the contribution 1 in the original manuscript (marked in red font).
>
> For weakness 2 (along with questions 3 and 4), we have added the routing results of NTHURouter, NCTU-GR and Labyrinth. We apologize for the missing of results of Labyrinth. Labyrinth fails to compile with recent versions of gcc; therefore, in the original submission we only reported the Labyrinth results that were already available in the HubRouter. We have now tried to refactor the Labyrinth codebase so that it can be built with the current toolchain and have rerun it on ISPD98 cases. The results of ISPD07 we reproduced seem to be unusable (WL across all benchmarks witness a significant decrease compared to the boderline) which we attribute to the incompatible input format of ISPD07. We are still trying to reproduce the complete results of ISPD07. The corresponding supplementary results have been added to the relevant table 3 and 4 in the revised manuscript. Regarding DGR, it is primarily designed as a detailed-routing solver, and its recently released open-source implementation is not directly applicable to global routing. We were unable to robustly adapt DGR to the global-routing setting within the timeframe of this revision, but we plan to continue working on this extension and will release the results once they become available.
>
> For weakness 3, we have revised our manuscript.
>
> For weakness 4 (along with questions 1 and 2), we have added the reasons of chosing of NTHURouter and NCTU-GR to the Appendix A.4.3 due to the page limit of the main text. Besides the complete ablation results are updated in the table 5.

---

### Official Review · Reviewer_XP4a · 2025-10-31

**Soundness:** 3
**Presentation:** 4
**Contribution:** 3
**Rating:** 6
**Confidence:** 5

**Summary:**

This paper introduces an end-to-end machine learning-based global router **ParetoRouter** for VLSI design, targeting the joint optimization of wirelength (WL) and overflow (OF) via multi-objective Pareto efficiency. Leveraging Flow Matching (FM) and a novel "Average Flow" loss, it integrates routing results from NTHURouter (OF-oriented) and NCTU-GR (WL-oriented) to generate diverse solutions. During inference, a one-step Das–Dennis sampling method enables controllable WL-OF trade-offs. Experiments show ParetoRouter achieves state-of-the-art OF reduction across benchmarks (e.g., ISPD98, ISPD07), maintains 100% connectivity, and achieves a 10× speedup compared to diffusion-based SOTA methods like DSBRouter.

**Strengths:**

This paper tries to address a critical gap in EDA by optimizing conflicting WL/OF objectives simultaneously—a practical necessity for industrial VLSI design. Its novelty lies in the first end-to-end ML-based multi-objective global router and the efficient Das–Dennis sampling for Pareto-front approximation. Writing is clear and the figures and tables are easy to understand, with well-structured contributions and thorough preliminaries. Besides, full code release is promised, while the architecture/training details are comprehensive.

**Weaknesses:**

1. The analysis of the loss function in the ablation study is relatively insufficient.
2. The data of DSBROUTER for IBM06 in Table 3 appears weird.
3. While the supplementary materials present actual routing results for cases of varied scales, they lack (providing these would significantly enhance reader comprehension):
   - Comparative analysis of the same cases against other baselines;
   - Explanations for loop occurrences observed in the results;
   - Case studies demonstrating the model's actual congestion avoidance capability.
4. The benchmarks appear outdated, would you consider adding experimental results on more recent ISPD 2018/2019 benchmarks?

**Questions:**

1. Table 3 shows higher inference times than NeuralSteiner/VAE-Hubrouter. What limits parallelization of the Das–Dennis sampling?
2. Training costs (GPU-hours) are omitted. Can you provide more training details?
3. Would you consider adding experimental results on more recent ISPD 2018/2019 benchmarks?

**Details Of Ethics Concerns:**

No ethical issues.

---

### Meta-Review · Area_Chair_bva1 · 2025-12-30

**Summary:**

This paper proposes an end-to-end machine learning-based global router ParetoRouter for VLSI design, jointly optimizing wirelength (WL) and overflow (OF) via multi-objective Pareto efficiency. Experimental results show the advantage of ParetoRouter over the SOTA ML-based router, DSBrouter.

However, reviewers indicated several major concerns, e.g., limited novelty of the proposed method, weak comparison with classic methods, and outdated benchmarks. The authors only provide responses to two reviewers with some explanations and limited additional experiments.

**Reviewer Concerns:**

Reviewer XP4a has major concerns: 1) insufficient analysis of the loss function in the ablation study; 2) weird data of DSBROUTER for IBM06 in Table 3; 3) more detailed analyses for some experimental results; 4) outdated benchmarks.

The authors did not provide responses to Reviewer XP4a.

Reviewer fSwF has major concerns: 1) Jointly optimizing overflow (OF) and wirelength (WL) is not very novel; 2) weak comparison with classic methods; 3) inaccurate survey of current works; 4) absence of a crucial ablation study.

The authors provided limited experiments in their response.

Reviewer Kuvv has major concerns: 1) missing important related works; 2) limited novelty (ParetoRouter vs. ParetoFlow) as most of the idea of the flow matching and the combination with Pareto Sampling have been proposed.

The authors provided responses to Reviewer XP4a, and mainly explained the difference between ParetoRouter and ParetoFlow.

Reviewer bZ4V has major concerns on generalization, complexity, and computational cost. But the authors did not provide responses.

**Reviewer Scores:**

The authors only provide responses to two reviewers with some explanations and limited additional experiments. Thus, I think most reviewers will not change their scores.

---

### Decision · Program_Chairs · 2026-01-26

Reject